# Sanity-Checking Pruning Methods:
# Random Tickets can Win the Jackpot

**Jingtong Su**[1,*]  **Yihang Chen**[2,*]  **Tianle Cai**[3,4,*]

**Tianhao Wu**[2]  **Ruiqi Gao**[3,4]  **Liwei Wang**[5,6,†]  **Jason D. Lee**[3,†]

[1]Yuanpei College, Peking University
[2]School of Mathematical Sciences, Peking University
[3]Department of Electrical Engineering, Princeton University
[4]Zhongguancun Haihua Institute for Frontier Information Technology
[5]Key Laboratory of Machine Perception, MOE, School of EECS, Peking University
[6]Center for Data Science, Peking University

## Abstract

Network pruning is a method for reducing test-time computational resource requirements with minimal performance degradation. Conventional wisdom of pruning algorithms suggests that: (1) Pruning methods exploit information from training data to find good subnetworks; (2) The architecture of the pruned network is crucial for good performance. In this paper, we conduct sanity checks for the above beliefs on several recent unstructured pruning methods and surprisingly find that: (1) A set of methods which aims to find good subnetworks of the randomly-initialized network (which we call "initial tickets"), hardly exploits any information from the training data; (2) For the pruned networks obtained by these methods, randomly changing the preserved weights in each layer, while keeping the total number of preserved weights unchanged per layer, does not affect the final performance. These findings inspire us to choose a series of simple *data-independent* prune ratios for each layer, and randomly prune each layer accordingly to get a subnetwork (which we call "random tickets"). Experimental results show that our zero-shot random tickets outperform or attain a similar performance compared to existing "initial tickets". In addition, we identify one existing pruning method that passes our sanity checks. We hybridize the ratios in our random ticket with this method and propose a new method called "hybrid tickets", which achieves further improvement.[3]

## 1  Introduction

Deep neural networks have achieved great success in the overparameterized regime [44, 35, 8, 7, 2]. However, overparameterization also leads to excessive computational and memory requirements. To mitigate this, network pruning [33, 22, 15, 14, 6] has been proposed as an effective technique to reduce the resource requirements with minimal performance degradation.

One typical line of pruning methods, exemplified by retraining [14], can be described as follows: First, find a subnetwork of original network using pruning methods (which we call the "pruning step"), and then, retrain this subnetwork to obtain the final pruned network (which we call the "retraining step").

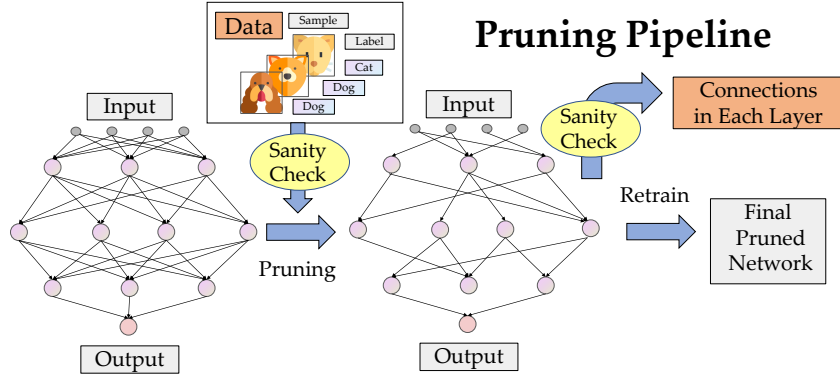

Figure 1: **A typical pipeline of obtaining the final pruned network.** The first step is a (usually) data-dependent pruning step that finds a subnetwork of the original network. The second step is a retraining step that trains the subnetwork to achieve minimal performance drop compared to the full network. Our sanity checks are applied to the data used in pruning step and the layerwise structure, i.e., connections in each layer, of the subnetwork obtained in the pruning step.

(Figure 1). Different pruning methods are mainly distinguished by *different criterion for finding the subnetwork*, *different weights to assign to the subnetwork* and *retraining scheme*.

Generally, there are two common beliefs behind this pruning procedure. First, it is believed that pruning methods *exploit the information from training data in the pruning step* to find good subnetworks. Therefore, there are only a few *data-independent* pruning methods [34], and it is not clear whether these methods can get similar (or even better) performance than the popular data-dependent pruning methods. Second, it is believed that *the architecture of the pruned network* is crucial for obtaining the final pruned network with good performance [14]. As a result, most of the papers in the pruning literature usually use "random pruning", i.e., randomly selecting some number of weights/neurons to be pruned according to the pruning ratio, as a weak baseline to show that the pruning methods outperform the trivial random architecture. However, we put into question whether these two beliefs really hold. To evaluate them, we propose two sets of sanity check approaches which examine whether the *data used in the pruning step* and *the structure of pruned network* are essential for the final performance of pruned networks for *unstructured* pruning methods (See Section 2.2) [4].

To evaluate the dependency on the data, we propose to corrupt the data *in the pruning step* while using the original data in the retraining step, and see if the subnetworks pruned using the corrupted data can achieve similar performance as those pruned using the true data. To test the importance of the architecture of the pruned subnetwork, we introduce a novel attack on the network architecture which we call *layerwise rearrange*. This attack rearranges all the connections of neurons *layerwise* and thus totally destroys the structure obtained in pruning step. It is worth noting that, after applying layerwise rearranging, the number of preserved weights in *each layer* stays unchanged. In contrast, if we rearrange the connections in the entire subnetwork, the resulting subnetwork is basically the same as that obtained by random pruning. We refer the readers to Figure 1 for an illustration of where we deploy our sanity checks.

We then apply our sanity checks on the pruning methods in four recent papers from ICLR 2019 and 2020 [23, 9, 41, 37]. We first classify the subnetworks found by these methods into "initial tickets", i.e., the weights before retraining are set to be the weights at initialization (this concept is the same as "winning tickets" in [9]), and "partially-trained tickets", i.e., the weights before retraining are set to be the weights from the middle of pretraining process[5] (See Section 4.1 for the formal definition and Figure 2 for an illustration). The effectiveness of "initial tickets" is highlighted by "the lottery ticket hypothesis (LTH)" [9], and the methods on "partially-trained tickets" make some modifications on the original method in [9].

We show that both of the beliefs mentioned above are not necessarily true for "initial tickets". Specifically, the subnetworks can be trained to achieve the same performance as initial tickets even

when they are obtained by using corrupted data in the pruning step or rearranging the initial tickets layerwise. This surprising finding indicates that true data may not be necessary for these methods and the architecture of the pruned network may only have limited impact on the final performance.

Inspired by the results of sanity checks, we propose to choose a series of simple *data-independent* pruning ratios for each layer, and randomly prune each layer accordingly to obtain the subnetworks (which we call "random tickets") at initialization. This zero-shot method pushes beyond the one-shot pruning-at-initialization methods [23, 41]. Concretely, our random tickets are drawn without any pretraining optimization or any data-dependent pruning criterion, and thus we save the computational cost of pruning. Experimental results show that our zero-shot random tickets outperforms or attains similar performance compared to all existing "initial tickets". Though we only try pruning ratios in a few simple forms without much tuning, the success of random tickets further suggests that our layerwise ratios can *serve as a compact search space for neural architecture search* [47, 48, 36, 25, 4].

In addition, we also find a very recent pruning method in ICLR 2020 [37] for obtaining "partially-trained tickets" can pass our sanity checks. We then hybridize the insights for designing random tickets with partially-trained tickets and propose a new method called "hybrid tickets". Experimental results show that the hybrid tickets can achieve further improve the partially-trained tickets.

In summary, our results advocate for a re-evaluation of existing pruning methods. Our sanity checks also inspire us to design data-independent and more efficient pruning algorithms.

## 2 Preliminaries

### 2.1 Notations

We use $\mathcal{D} = \{\mathbf{x}_i, y_i\}_{i=1}^n$ to denote the training dataset where $\mathbf{x}_i$ represents the sample and $y_i$ represents the label. We consider a neural network with $L$ layers for classification task on $\mathcal{D}$. We use vector $\mathbf{w}_l \in \mathbb{R}^{m_l}$ to denote the weights in layer $l$ where $m_l$ is the number of weights in layer $l$. Then the network can be represented as $f(\mathbf{w}_1, \cdots, \mathbf{w}_L; \mathbf{x})$ where $\mathbf{x}$ is the input to the network. The goal of classification is minimizing an error function $\ell$ over the outputs of network and target labels, i.e., $\sum_{i=1}^n \ell\left(f(\mathbf{w}_1, \cdots, \mathbf{w}_L, \mathbf{x}_i), y_i\right)$, and the performance is measured on a test set.

### 2.2 Unstructured Pruning (Pruning Individual Weights)

One major branch of network pruning methods is unstructured pruning, and it dates back to Optimal Brain Damage [22] and Optimal Brain Surgeon [16], which prune weights based on the Hessian of the loss function. Recently, [13] proposes to prune network weights with small magnitude and incorporates the pruning method to "Deep Compression" pipeline to obtain efficient models. Several other criterions and settings are proposed for unstructured pruning[6, 39, 31]. Apart from unstructured pruning, there are structured pruning methods [20, 29, 26, 21, 24, 42, 1] which prune at the levels of convolution channels or higher granularity. In this paper, we mainly focus on *unstructured pruning*.

Unstructured pruning can be viewed as a process to find a mask on the weights which determines the weights to be preserved. Formally, we give the following definitions for a clear description:

**Mask.** We use $\mathbf{c}_l \in \{0, 1\}^{m_l}$ to denote a mask on $\mathbf{w}_l$. Then the pruned network is given by $f(\mathbf{c}_1 \odot \mathbf{w}_1, \cdots, \mathbf{c}_L \odot \mathbf{w}_L; \mathbf{x})$.

**Sparsity and keep-ratios.** The sparsity $p$ of the pruned network calculates as $p = 1 - \frac{\sum_{l=1}^L \|\mathbf{c}_l\|_0}{\sum_{l=1}^L m_l}$, where $\|\cdot\|_0$ denotes the number of nonzero elements. We further denote $p_l$ as the "keep-ratio" of layer $l$ calculated as $\frac{\|\mathbf{c}_l\|_0}{m_l}$.

### 2.3 Pruning Pipeline and Tickets

In Figure 1, we show a typical pipeline of network pruning in which we first prune the full network and obtain a subnetwork, then retrain the pruned subnetwork. The processes of pruning and retraining can be repeated iteratively [13], which requires much more time. In this paper, we focus on *one shot pruning* [14], which does not iterate the pipeline.

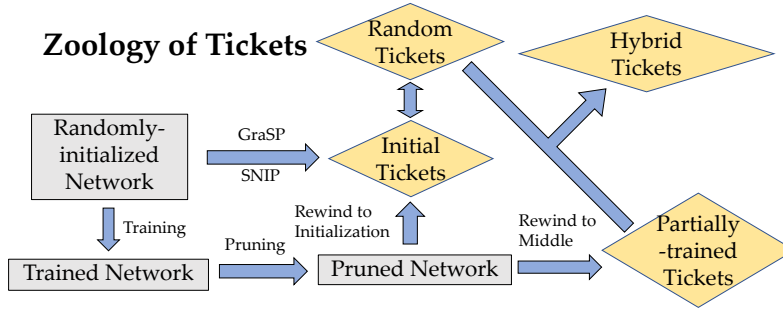

Figure 2: **Zoology of the tickets considered in this paper.** Initial tickets and partially trained tickets are introduced in Section 2.3, random tickets are proposed in Section 4.2, and hybrid tickets are proposed in Section 5.2.

We evaluate several recently proposed pruning methods which can be classified as follows using the names "tickets" adopted from "Lottery Ticket Hypothesis" [9]:

- **Initial tickets**: This kind of methods aim to find a subnetwork of the randomly-initialized network which we call "*initial tickets*" that can be trained to reach similar test performance of the original network. Initial tickets include pruning at initialization [23, 41] and the method proposed in [9].

- **Partially-trained tickets**: Different from initial tickets, partially-trained tickets are constructed by first training the network, pruning it, and then *rewinding* the weights to some middle stage [37]. Initial tickets found by the method in [9] can be viewed as rewinding to epoch 0.

We illustrate these two kinds of tickets in Figure 2 together with our proposed new tickets that will be introduced in later sections.

## 3 Sanity Check Approaches

In this section, we propose sanity check approaches for testing if some common beliefs of pruning methods truly hold in several recently proposed methods.

### 3.1 Checking the Dependency on Data

Popular pruning methods usually utilize data in the pruning step to find a good pruned subnetwork that can be trained to perform as well as the full network on the same data (Figure 1).

However, there is no clear evidence showing that the data used in pruning step is important. Furthermore, from the few existing data-independent pruning methods [34], it is also hard to judge whether data-independent methods have the potential to attain similar performance as popular data-dependent pruning methods. Therefore, we ask the following question:

*Is data information useful for finding good subnetworks in the pruning step?*

To answer this question, we introduce two operations, using random labels and random pixels, to corrupt the data [44]. Note that different from [44] in which these two operations are used to demonstrate whether overparameterized network can converge on bad data, we apply these operations on the data used in the *pruning step* to check *if corrupted data still leads to good subnetworks*. In addition to these strong corruptions, we also provide a weaker check that only reduces the number of data but not corrupts the data. Concretely, we define the two operations as follows:

- **Random labels**: In the pruning step, we replace the dataset $\mathcal{D} = \{\mathbf{x}_i, y_i\}_{i=1}^n$ with $\hat{\mathcal{D}} = \{\mathbf{x}_i, \hat{y}_i\}_{i=1}^n$ where $\hat{y}_i$ is randomly generated from a uniform distribution over each class.

- **Random pixels**: In the pruning step, we replace the dataset $\mathcal{D} = \{\mathbf{x}_i, y_i\}_{i=1}^n$ with $\hat{\mathcal{D}} = \{\hat{\mathbf{x}}_i, y_i\}_{i=1}^n$ where $\hat{\mathbf{x}}_i$ is the randomly shuffled[6] $\mathbf{x}_i$ and shuffling of each individual sample is independent.

- **Half dataset**: We reduce the number of data used in the pruning step. For convenience, in this setting, we simply take half of the data in the dataset for the pruning step.

## 3.2 Checking the Importance of the Pruned Network's Architecture

In unstructured pruning, the architecture, or the connections of neurons, of the pruned subnetwork has long been considered as the key to make the pruned network able to achieve comparable performance to the original network [14]. As a result, random pruning, i.e., randomly selecting $(1 - p) \cdot \sum_{l=1}^{L} m_l$ weights to preserve according to a given sparsity level $p$, is widely used as a baseline method to show the effectiveness of pruning methods with more carefully designed criterion.

However, we cast doubt on whether these individual connections in the pruned subnetwork is crucial, or if there any intermediate factors that determine the performance of pruned network. Concretely, we answer the following question:

*To what extent does the architecture of the pruned subnetwork affect the final performance?*

Towards answering this question, we propose a novel layerwise rearranging strategy, which keeps the number of preserved weights in each layer but *completely destroys the network architecture (connections) found by the pruning methods* for each individual layer. As an additional reference, we also introduce a much weaker layerwise weight shuffling operation, which only shuffle the unmasked weights but keep the connections.

- **Layerwise rearrange**: After obtaining the pruned network, i.e., $f(\mathbf{c}_1 \odot \mathbf{w}_1, \cdots, \mathbf{c}_L \odot \mathbf{w}_L, \mathbf{x})$, we randomly rearrange the mask $\mathbf{c}_l$ of *each layer, independently,* into $\hat{\mathbf{c}}_l$, and then train on the network with the rearranged masks $f(\hat{\mathbf{c}}_1 \odot \mathbf{w}_1, \cdots, \hat{\mathbf{c}}_L \odot \mathbf{w}_L; \mathbf{x})$. We give an illustration of this operation in Appendix A.
- **Layerwise weights shuffling**: Layerwise weights does not change the masks but shuffles the *unmasked weights* in each layer independently.

With the approaches above, we are ready to perform sanity-check on existing pruning methods.

## 4 Case Study: Initial Tickets

In this section, we will apply our sanity checks to "initial tickets" defined in Section 2.3. Surprisingly, our results suggest that the final performance of the retrained "initial tickets" does not drop when using corrupted data including random labels and random pixels in the pruning step. Moreover, the layerwise rearrangement does not affect the final performance of "initial tickets" either. This finding further inspires us to design a *zero-shot* pruning method, dubbed "random tickets".

### 4.1 Initial Tickets Fail to Pass Sanity Checks

We conduct experiments on three recently proposed pruning methods that can be classified as "initial tickets". We first briefly describe these methods:

- **SNIP [23] and GraSP [41]**: These two methods prune the network at initialization by finding the mask using different criterion. Specifically, SNIP leverages the notion of connection sensitivity as a measure of importance of the weights to the loss, and then remove unimportant connections. GraSP aims to preserve the gradient flow through the network by maximizing the gradient of the pruned subnetwork. The merit of these methods is that they can save the time of training by pruning at initialization compared to methods that require pretraining[13].
- **Lottery Tickets (LT) [9]**: The pruning procedure of Lottery Tickets follows a standard three-step-pipeline: training a full network firstly, using magnitude-based pruning [13] to obtain the ticket's architecture, and *resetting the weights to the initialization* to obtain the initial ticket.[7]

After obtaining the initial tickets, we train them by standard methods and evaluate the test accuracy of the final models. We follow a standard training procedure as [23, 41, 17] *throughout the entire paper* on ResNet [19] and VGG [38]. The detailed setting can be found in Appendix B.

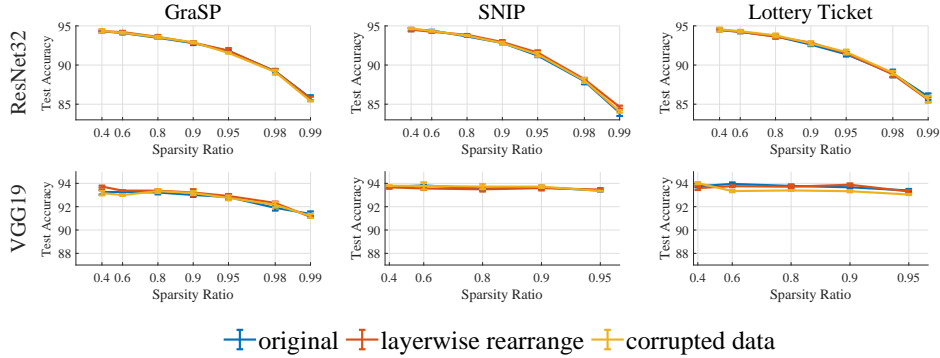

Figure 3: Sanity checks of the initial tickets of ResNet32 and VGG19 on CIFAR10.

To sanity-check these three methods and answer the motivating questions proposed in Section 3, we apply the two strongest checks in our toolbox.

We first check if the pruning procedure can utilize information on the training data. Towards this goal, we use *the combination of random label and random pixel corruptions* in Section 3 in the pruning step. We then train these tickets on the *original CIFAR-10 dataset*. Surprisingly, all initial tickets generated by the corrupted dataset behave as well as the originally picked tickets (Figure 3).[8] That is to say, *even corrupted data can be used to find good initial tickets*.

Given the weak dependency on the data used in the pruning step, we then investigate whether the architecture (connections between neurons) of the initial tickets matter. We then apply our *layerwise rearrange* attack, which totally removes the dependency of the learned connections in each layer on the initial tickets. We find that the performance of initial tickets does not drop when applying the layerwise rearrange attack (Figure 3). This contradicts the belief that the pruned structure is essential for obtaining a subnetwork with good performance.[9] We also conduct experiments on the CIFAR-100 dataset. The results are deferred to Appendix **??**.

## 4.2 Random Tickets Win the Jackpot

The results of sanity check on initial tickets suggest that:

- Data may not be important for finding good initial tickets.

- The connections of neurons in individual layers in initial tickets can be totally rearranged without any performance drop, indicating that only the number of remained connections or keep-ratio (recall the definition in Section 2.2) in each layer matters.

Based on these observations, we only need to find a series of good *keep-ratios* and randomly prune the network accordingly. We call the subnetworks obtained by this rule "random tickets". As illustrated in the Ticket Zoo in Figure 2, random tickets can be viewed as a special kind of initial tickets since they use the weights at initialization. Meanwhile, applying layerwise rearrangement to initial tickets obtained by existing pruning methods can transform these initial tickets to random tickets with the layerwise keep-ratios determined by the initial tickets.

Though we can obtain the keep-ratios from existing initial tickets in the above way, we are more willing to push even further to directly design a series of *simpler but effective* keep-ratios. Towards this goal, we first investigate the ratios of the initial tickets obtained using GraSP, SNIP and Lottery Tickets on both VGG and ResNet architectures. From these initial tickets, we extract the following principle of keep-ratios that leads to good final performance:

Table 1: Test accuracy of pruned VGG19 and ResNet32 on CIFAR-10 and CIFAR-100 datasets. In the full paper, the bold number indicates the average accuracy is within the best confidence interval.

| Dataset | CIFAR-10 | | | CIFAR-100 | | |
|---|---|---|---|---|---|---|
| Pruning ratio | 90% | 95% | 98% | 90% | 95% | 98% |
| **VGG19** (Full Network) | 93.70 | - | - | 72.60 | - | - |
| LT [9] | 93.66±0.08 | **93.39±0.13** | *10.00±0.00* | 72.58±0.27 | 70.47±0.19 | *1.00±0.00* |
| SNIP [23] | 93.65±0.14 | **93.39±0.08** | 81.37±18.17 | **72.83±0.16** | **71.81±0.11** | 10.83±6.74 |
| GraSP [41] | 93.01±0.16 | 92.82±0.20 | 91.90±0.23 | 71.07±0.31 | 70.14±0.21 | 68.34±0.20 |
| Random Tickets (Ours) | **93.77±0.10** | **93.42±0.22** | **92.45±0.22** | 72.55±0.14 | 71.37±0.09 | **68.98±0.34** |
| **ResNet32** (Full Network) | 94.62 | - | - | 74.57 | - | - |
| LT [9] | 92.61±0.19 | 91.37±0.28 | 88.92±0.49 | 69.63±0.26 | 66.48±0.13 | **60.22±0.60** |
| SNIP [23] | 92.81±0.17 | 91.20±0.19 | 87.94±0.40 | 69.97±0.17 | 64.81±0.44 | 47.97±0.82 |
| GraSP [41] | 92.79±0.24 | **91.80±0.11** | **89.21±0.26** | **70.12±0.15** | **67.05±0.39** | 59.25±0.33 |
| Random Tickets (Ours) | **92.97±0.05** | 91.60±0.26 | **89.10±0.33** | 69.70±0.48 | 66.82±0.12 | **60.11±0.16** |

Table 2: Test accuracy of pruned VGG19 and ResNet32 on Tiny-Imagenet dataset.

| Network | VGG19 | | | ResNet32 | | |
|---|---|---|---|---|---|---|
| Pruning ratio | 90% | 95% | 98% | 90% | 95% | 98% |
| **VGG19/ResNet32** (Full Network) | 61.57 | - | - | 62.92 | - | - |
| SNIP [23] | **61.65±0.17** | 58.72±0.20 | 9.37±13.67 | 54.53±0.34 | 47.68±0.20 | 32.09±1.87 |
| GraSP [41] | 60.71±0.25 | **59.11±0.03** | **57.08±0.16** | **56.23±0.09** | 51.36±0.37 | 39.43±1.52 |
| Random Tickets (Ours) | 60.94±0.38 | **59.48±0.43** | 55.87±0.16 | 55.26±0.22 | **51.41±0.38** | **43.93±0.18** |

**Keep-ratios decrease with depth.** For all these methods on both VGG and ResNet, we observe that the keep-ratios of initial tickets have the common behavior that the keep-ratio of a deeper layer is lower, i.e., the keep-ratios decay as a function of depth. Moreover, the keep-ratios of VGG decays faster than ResNet for all three algorithms.[10]

We also observe that apart from the overall trend of declining in layerwise keep-ratios, the keep-ratios in some special layers behave differently. For example, the keep-ratio of the downsampling layers in ResNet are significantly larger than their neighboring layers.

**Smart ratios.** Based on these observations, we propose *smart-ratios*, a series of keep-ratios that take the form of simple decreasing functions. Specifically, for an $L$-layer ResNet, we set the keep-ratio of the linear layer to 0.3 for any sparsity $p$, and let the keep-ratio of a given layer $l$, i.e., $p_l$ (Section 2.2), to be proportional to $(L - l + 1)^2 + (L - l + 1)$. For VGG, we divide $(L - l + 1)^2 + (L - l + 1)$ by $l^2$ for a faster decrease. The details can be found in Appendix C.

We note that we choose our ratio just by trying a few decreasing functions *without careful tuning*, but already get good performance. We also conduct ablation studies on ascending keep-ratios and different rates of decaying keep-ratios. Experimental results suggest that the descending order and, upon that, the descent rate of keep-ratios is crucial for the final performance of random tickets. This partially explains why the baseline of randomly pruning over the whole network other than using a tailored layerwise ratio usually performs poorly. Due to space limitation, we put the detailed ablation study in the Appendix. Moreover, we point out that as good keep-ratios produce good subnetworks, one may use keep-ratios as a compact search space for Neural Architecture Search, which may be an interesting future direction.

The test accuracy on CIFAR-10/CIFAR-100 and Tiny-Imagenet datasets are shown in Table 1 and 2. As reported, our random tickets with smart ratios surpass several carefully designed methods on several benchmarks and sparsity. And we note that when the sparsity is very high, some layers might retain only few weights when using SNIP or LT, so that the test accuracy drops significantly [17]. Recently [40] also observe this phenomenon and propose a pruning method without data and be able to avoid the whole-layer collapse.

# 5   Case Study: Partially-trained Tickets

In this section, we study pruning methods in a very recent ICLR 2020 paper [37], which is classified as partially-trained tickets (Section 2.3). These methods can pass our sanity checks on data dependency and architecture. We then combine our insights of random tickets with these partially-trained tickets and propose a method called "hybrid tickets" (Figure 2) which further improves upon [37].

## 5.1   Partially-trained Tickets Pass Sanity Checks

Different from initial tickets, [37] propose a *learning rate rewinding* method that improves beyond *weights rewinding* [10]. The methods used in [37] include weights rewinding and learning rate rewinding. Both methods first fully train the unpruned network (which we call pretraining) and generate the mask by magnitude pruning [14]. Then the two methods for retraining are

- **Weights rewinding**: First rewind the unmasked weights of the pruned subnetwork to their values at some middle epoch of the pretraining step and then retrain the subnetwork using the same training schedule as the pretraining step at that epoch.
- **Learning rate rewinding**: Retrain the masked network with its learned weights using the original learning rate schedule.

These two methods can be classified as "partially-trained tickets" as they use weights from partially-trained (or actually for the latter, fully-trained) network. And as pointed out in [37], learning rate rewinding usually surpasses weights rewinding, so we mostly focus on learning rate rewinding.

Similar to Section 4.1, we conduct sanity checks on learning rate rewinding method. We show the results of two weaker attacks, layerwise weight shuffling and half dataset (see Section 3) since if learning rate rewinding can pass these checks, i.e., its performance degrades under these weaker attacks, it will naturally pass the previous sanity check with strong attacks used in Section 4.1.

As shown in Table 3, both modifications do impair the performance. We note that the degradation of test accuracy on VGG is not such significant as that on ResNet, which is understandable since the number of parameters of VGG is huge and the gaps between different methods are small. When using layerwise weight shuffling, the test accuracy is reduced to being comparable to the case when the weights are rewound to the initialization (LT). When using half dataset in the pruning step, the test accuracy also drops compared to learning rate rewinding while it is higher than LT (Section 4.1). We conclude from these observations that the partially-trained tickets can pass our sanity checks.

## 5.2   Hybrid Tickets

As the results of the sanity checks on learning rate rewinding suggest, this method truly encodes information of data into the weights, and the architectures of the pruned subnetworks cannot be randomly changed without performance drop.

On another side, the success of random tickets shed light on using smarter keep-ratios to attain better performance. Therefore, we combine our "smart ratios" (Section 4.2) and the idea of learning rate rewinding. Concretely, we first pretrain the network as in learning rate rewinding, and then, while pruning the pretrained network, we use magnitude-based pruning *layerwise* and the keep-ratios are determined by the *smart-ratios*. In other words, for a given layer $l$, we only keep those weights whose magnitude are in the largest $p_l$ portion in layer $l$, where $p_l$ is the smart-ratio of layer $l$. And finally, the pruned network is retrained with a full learning-rate schedule as learning rate rewinding does.

We call the pruned subnetworks obtained by this hybrid method "hybrid tickets" (See Figure 2). The test accuracy on CIFAR-10 and CIFAR-100 is shown in Table 4. As reported, our hybrid tickets gain further improvement upon learning rate rewinding, especially at high sparsity for both datasets and network architectures. At the same time, hybrid tickets can avoid pruning the whole layer [17] which may happen for learning rate rewinding.

# 6   Conclusion and Discussion

In this paper, we propose several sanity check methods (Section 3) on unstructured pruning methods that test whether the data used in the pruning step and whether the architecture of the pruned

Table 3: Sanity-check on partially-trained tickets on CIFAR-10 dataset.

| Network | VGG19 | | | ResNet32 | | |
|---|---|---|---|---|---|---|
| Pruning ratio | 90% | 95% | 98% | 90% | 95% | 98% |
| **VGG19/ResNet32** (Full Network) | 93.70 | - | - | 94.62 | - | - |
| LT [9] | 93.66±0.08 | 93.39±0.13 | *10.00±0.00* | 92.61±0.19 | 91.37±0.28 | 88.92±0.49 |
| Shuffle Weights | 93.54±0.04 | 93.33±0.10 | *10.00±0.00* | 92.38±0.36 | 91.29±0.28 | 88.52±0.47 |
| Half Dataset | 93.79±0.14 | 93.53±0.13 | *10.00±0.00* | 93.01±0.18 | 92.03±0.21 | 89.95±0.08 |
| Learning Rate Rewinding [37] | **94.14±0.17** | **93.99±0.15** | *10.00±0.00* | **94.14±0.10** | **93.02±0.28** | **90.83±0.22** |

Table 4: Test accuracy of partially-trained tickets and our hybrid tickets of VGG19 and ResNet32 on CIFAR-10 and CIFAR-100 datasets.

| Dataset | CIFAR-10 | | | CIFAR-100 | | |
|---|---|---|---|---|---|---|
| Pruning ratio | 90% | 95% | 98% | 90% | 95% | 98% |
| **VGG19** (Full Network) | 93.70 | - | - | 72.60 | - | - |
| OBD [22] | 93.74 | 93.58 | 93.49 | 73.83 | 71.98 | 67.79 |
| Learning Rate Rewinding [37] | **94.14±0.17** | **93.99±0.15** | *10.00±0.00* | **73.73±0.18** | 72.39±0.40 | *1.00±0.00* |
| Hybrid Tickets (Ours) | **94.00±0.12** | 93.83±0.10 | **93.52±0.28** | 73.53±0.20 | **73.10±0.11** | **71.61±0.46** |
| **ResNet32** (Full Network) | 94.62 | - | - | 74.57 | - | - |
| OBD [22] | 94.17 | 93.29 | 90.31 | 71.96 | 68.73 | 60.65 |
| Learning Rate Rewinding [37] | **94.14±0.10** | **93.02±0.28** | **90.83±0.22** | **72.41±0.49** | 67.22±3.42 | 59.22±1.15 |
| Hybrid Tickets (Ours) | 93.98±0.15 | 92.96±0.13 | **90.85±0.06** | 71.47±0.26 | **69.28±0.40** | **63.44±0.34** |

subnetwork are essential for the final performance. We find that one kind of pruning method, classified as "initial tickets" (Section 2.3) hardly exploit any information from data, because randomly changing the preserved weights of the subnetwork obtained by these methods layerwise does not affect the final performance. These findings inspire us to design a *zero-shot data-independent* pruning method called "random tickets" which outperforms or attains similar performance compared to initial tickets. We also identify one existing pruning method that passes our sanity checks, and hybridize the random tickets with this method to propose a new method called "hybrid tickets", which achieves further improvement. Our findings bring new insights in rethinking the key factors on the success of pruning algorithms.

Besides, a concurrent and independent work [11] got a similar conclusion to our layerwise rearrange sanity check. As a complementary to our results, the results on ImageNet in [11] shows our finding can be generalized to large-scale datasets. Both our works advocate to rigorously sanity-check future pruning methods and take a closer look at opportunities to prune early in training.

## Acknowledge

We thank Jonathan Frankle, Mingjie Sun and Guodong Zhang for discussions on pruning literature. TC and RG are supported in part by the Zhongguancun Haihua Institute for Frontier Information Technology. LW was supported by National Key R&D Program of China (2018YFB1402600), Key-Area Research and Development Program of Guangdong Province (No. 2019B121204008) and Beijing Academy of Artificial Intelligence. JDL acknowledges support of the ARO under MURI Award W911NF-11-1-0303, the Sloan Research Fellowship, and NSF CCF 2002272.

## Broader Impact

We investigate into neural network pruning methods, which is an important way to reduce computational resource required in modern deep learning. It can potentially help enable faster inference, conserve less energy, and make AI widely deployable and assessable in mobile and embedded systems. Our experiments can also provide more insight for neural networks in deep learning algorithms, which could potentially lead to the development of better deep learning algorithms.

## Footnotes

*Equal Contribution, reverse alphabetical order.

[3]Our code is publicly available at `https://github.com/JingtongSu/sanity-checking-pruning`.

[4]Due to space limitation, we review some related works in Appendix **??**.

[5]When clear from the context, we may use "tickets" to refer to the subnetworks obtained by certain methods or directly refer to the methods.

[6]By shuffling, we mean generating a random permutation and then reordering the vector by this permutation.

[7]In the main body of our paper, we focus on one-shot pruning methods since: 1. SNIP and GraSP are both pruning methods without significant training expenses, and 2. iterative pruning cannot consistently outperform one shot pruning, and even when iterative pruning is better, the gap is small, as shown in [27] and [37]. The discussions and experiments about Iterative Magnitude Pruning are deferred to Appendix **??**, from which we conclude the iterative procedure hardly helps initial tickets.

[8]We omit results with significantly high variances or are trivial, i.e. have a 10% accuracy. This kind of omitting is used on figures shown throughout our paper.

[9]In the paper [32]'s Appendix A2 part, the authors report applying layerwise rearrange can hurt the performance when the pruning ratio is high. However, in their setting they use a "late resetting" trick which is kind of similar to the Learning Rate Rewinding as we will discuss in the next section of our paper. So it is not surprising to see that Figure A1 in [32] show evidence that the method may pass the layerwise rearrange check.

[10]Similar trends are also reported in Figure 3 of [41], Figure 1b of [32] and Figure 11 of [11].

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
