[Supplementary Material]

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

. YC and TW are partially supported by the elite undergraduate training program of School of Mathematical Sciences in Peking University. TC and RG are supported in part by the Zhongguancun Haihua Institute for Frontier Information Technology. LW was supported by National Key R&D Program of China (2018YFB1402600), Key-Area Research and Development Program of Guangdong Province (No. 2019B121204008) and Beijing Academy of Artificial Intelligence. JDL acknowledges support of the ARO under MURI Award W911NF-11-1-0303, the Sloan Research Fellowship, and NSF CCF 2002272.

## Broader Impact

We investigate into neural network pruning methods, which is an important way to reduce computational resource required in modern deep learning. It can potentially help enable faster inference, conserve less energy, and make AI widely deployable and assessable in mobile and embedded systems. Our experiments can also provide more insight for neural networks in deep learning algorithms, which could potentially lead to the development of better deep learning algorithms.

## Footnotes

*Equal Contribution, reverse alphabetical order.

[2]Our code is publicly available at `https://github.com/JingtongSu/sanity-checking-pruning`.

[3]Due to space limitation, we review some related works in Appendix F.

[4]When clear from the context, we may use "tickets" to refer to the subnetworks obtained by certain methods or directly refer to the methods.

[5]By shuffling, we mean generating a random permutation and then reordering the vector by this permutation.

[6]In the main body of our paper, we focus on one-shot pruning methods since: 1. SNIP and GraSP are both pruning methods without significant training expenses, and 2. iterative pruning cannot consistently outperform one shot pruning, and even when iterative pruning is better, the gap is small, as shown in [27] and [37]. The discussions and experiments about Iterative Magnitude Pruning are deferred to Appendix D, from which we conclude the iterative procedure hardly helps initial tickets.

[7]We omit results with significantly high variances or are trivial, i.e. have a 10% accuracy. This kind of omitting is used on figures shown throughout our paper.

[8]In the paper [32]'s Appendix A2 part, the authors report applying layerwise rearrange can hurt the performance when the pruning ratio is high. However, in their setting they use a "late resetting" trick which is kind of similar to the Learning Rate Rewinding as we will discuss in the next section of our paper. So it is not surprising to see that Figure A1 in [32] show evidence that the method may pass the layerwise rearrange check.

[9]Similar trends are also reported in Figure 3 of [41], Figure 1b of [32] and Figure 11 of [11].

[10]Can be downloaded from `http://cs231n.stanford.edu/tiny-imagenet-200.zip`

[11] 30% is similar to those keep-ratios of linear layers obtained by SNIP and GraSP, and we also find that the choice of keep-ratio of linear layer hardly affects the final performance.

[12] $L$ denotes the total number of convolutional and linear layers in a network.

[13] As a comparison, the original VGGIMP's 0.98 sparsity result is $78.85 \pm 9.76$.

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

# A Illustration of Layerwise Rearrange

The operation layerwise arrangement is illustrated in Figure 4.

Figure 4: **Layerwise rearrange.** This operation keeps the number of preserved weights in each layer but totally destroys the network architecture (connections) found by the pruning methods for each individual layer.

# B Experiment Settings

In all experiments throughout the paper, we use three benchmark image classification datasets, i.e., CIFAR-10, CIFAR-100 and Tiny-ImageNet[10] with VGG [38] and ResNet [19]. The CIFAR-10 dataset consists of 50,000 32x32 color (three-channel) training examples and 10,000 test examples, and has 10 classes of labels. The CIFAR-100 dataset consists of 50,000 32x32 color training examples and 10,000 test examples. Different from CIFAR-10, CIFAR-100 has 100 classes of labels with 500/100 training/test examples for each class. The Tiny-Imagenet dataset consists of 100,000 64x64 color training examples and 10,000 test examples. It has 200 classes, and each class has 500 training examples with 50 validation examples.

Following the setting in [41, 17], we double the number of filters in each convolutional layer of all the ResNet architectures throughout the paper in order to make comparisons with these baseline algorithms. The pruned network is trained with Kaiming initialization [18], using SGD for 160 epochs for CIFAR-10/100, and 300 epochs for Tiny-ImageNet, with an initial learning rate of 0.1 and batch size 64. The learning rate is decayed by a factor of 0.1 at 1/2 and 3/4 of the total number of epochs, together with a weight decay factor of 1e-4. Moreover, we run each experiment for 3 trials and report the best test accuracy and the corresponding standard deviations.

Our code is based on the released code of [41, 27] (`https://github.com/alecwangcq/GraSP`, `https://github.com/Eric-mingjie/rethinking-network-pruning`).

# C Details of Random Tickets and Ablation Studies

In this section, we first describe the detailed process of generating random tickets. Then we provide ablation studies on the performance of different smart ratios; and the performance of our proposed methods (random tickets, hybrid tickets) on different network architectures.

## C.1 Generation of Random Tickets

We first note that in different pruning methods, the number of weights retained in linear layer is chosen differently. In SNIP [23] and GraSP [41], the linear layer is pruned together with other

Table 5: Ablation study of different keep-ratios on CIFAR-10 dataset.

| Network | VGG19 | | | ResNet32 | | |
|---|---|---|---|---|---|---|
| Pruning ratio | 90% | 95% | 98% | 90% | 95% | 98% |
| **VGG19/ResNet32** (Full Network) | 93.70 | - | - | 94.62 | - | - |
| Random Tickets (Ours) | **93.77±0.10** | **93.42±0.22** | **92.45±0.22** | **92.97±0.05** | **91.60±0.26** | **89.10±0.33** |
| Balanced | 91.30±0.22 | 89.79±0.22 | 86.31±0.32 | 91.78±0.17 | 90.06±0.16 | 86.52±0.36 |
| Ascending | 91.79±0.24 | 90.29±0.17 | 87.22±0.30 | *10.00±0.00* | *10.00±0.00* | *10.00±0.00* |
| Linear Decay | **93.83±0.23** | **93.34±0.12** | 92.32±0.14 | 92.68±0.03 | **91.60±0.23** | 88.54±0.14 |
| Cubic Decay | **93.90±0.15** | **93.63±0.29** | **92.66±0.28** | 92.74±0.13 | **91.56±0.23** | **89.14±0.16** |

convolutional layers, while in Lottery Tickets [9], the linear layer is fully preserved. For simplicity, we fix the keep-ratio of linear layer to be 30%[11] for all experiments on smart ratio.

The smart ratios for the convolutional layers are designed as follows. As mentioned in Section 4.2, for an $L$-layer ResNet[12], we set the keep-ratio of $l$-th layer to be proportional to $(L-l+1)^2+(L-l+1)$. Concretely, this is achieved by following steps:

1. Calculate the number of weights to be retained as $(1-p)\sum_{i=1}^{L} m_l$.

2. Set the keep-ratio of layer $l$ as $(L-l+1)^2+(L-l+1)$.

3. Linearly scale the keep-ratio of each layer such that the final number of retained weights is equal to $(1-p)\sum_{i=1}^{L} m_l$.

For an $L$-layer VGG net, the process is similar except that we add an extra heterogeneous constant penalty for each layer to make the smart ratios decay faster. Specifically, we replace $(L-l+1)^2+(L-l+1)$ with $\frac{(L-l+1)^2+(L-l+1)}{l^2}$ in the second step.

We further note that, for several small sparsity parameters $p$ (e.g., $p=0.9$), linearly scaling the keep-ratio as in step 3 may result in a keep-ratio greater than 1. Under these circumstances, we simply set these layers' keep-ratios to be 1 and move the extra retained parameters to the deeper layer immediately behind them.

## C.2 Ablation Studies on Smart Ratio

We conduct experiments on CIFAR-10 dataset to compare several types of keep-ratios including ascending, balanced, and descending.

1. **Ascending keep-ratio**: Reverse our smart ratio.

2. **Balanced keep-ratio**: Set the keep-ratio of each layer to be $1-p$, where $p$ is the target sparsity.

3. **Linear decay**: Set the keep-ratio of $l$-th convolutional layer to be proportional to $L-l+1$.

4. **Cubic decay**: Set the keep-ratio of $l$-th convolutional layer to be proportional to $(L-l+1)^3$.

For all the ratios, the keep-ratio of linear layer is uniformly set to be 30%. The results can be found in Table 5. We observe that the balanced and ascending keep-ratios lead to much worse performance than the descending keep-ratios, while descending keep-ratios with different decaying speeds are comparative on CIFAR10. As mentioned in Section 4.2, we did not tune the ratio so much, and the results in Table 5 suggest that to get a better performance, one may use different keep-ratios for different sparsity to generate high-performance random tickets.

## C.3 Ablation Studies on Different Architectures

We test our proposed random tickets and hybrid tickets on more architectures including ResNet20/56 and VGG11/16 on CIFAR-10/100 datasets. The comparison of our random tickets and those baseline methods with VGG11/16 and ResNet20/56 on CIFAR-10/100 can be found in Table 6, while the results of our hybrid tickets can be found in Table 7. These results show that our smart ratio can generalize at both ResNet/VGG architectures with different depths.

Table 6: Test accuracy of pruned VGG11/16 and ResNet20/56 on CIFAR-10 and CIFAR-100 datasets.

| Dataset | CIFAR-10 | | | CIFAR-100 | | |
|---|---|---|---|---|---|---|
| Pruning ratio | 90% | 95% | 98% | 90% | 95% | 98% |
| **VGG11** (Full Network) | 92.05 | - | - | 69.09 | - | - |
| LT [9] | **91.50±0.27** | 90.35±0.12 | 86.71±0.34 | **68.58±0.18** | 67.42±0.32 | 65.23±0.55 |
| SNIP [23] | 91.29±0.06 | **90.77±0.35** | **89.16±0.07** | 68.65±0.58 | 67.48±0.35 | 64.05±0.37 |
| GraSP [41] | 90.48±0.08 | 89.70±0.12 | 88.46±0.31 | 66.11±0.18 | 64.47±0.56 | 62.07±0.18 |
| Random Tickets (Ours) | 91.67±0.24 | 90.86±0.26 | 89.11±0.16 | 68.82±0.47 | 66.90±0.09 | 63.85±0.19 |
| **VGG16** (Full Network) | 93.77 | - | - | 73.24 | - | - |
| LT [9] | 93.56±0.17 | **93.23±0.08** | *36.27±45.5* | **72.32±0.11** | 70.65±0.18 | 64.88±0.72 |
| SNIP [23] | 93.47±0.16 | 93.13±0.05 | **92.05±0.02** | 72.41±0.18 | **71.18±0.26** | 67.25±0.37 |
| GraSP [41] | 93.03±0.16 | 92.71±0.33 | 91.79±0.11 | 71.08±0.43 | 69.99±0.41 | 67.68±0.29 |
| Random Tickets (Ours) | **93.80±0.06** | **93.31±0.17** | 92.13±0.15 | 72.42±0.14 | 71.12±0.28 | 68.17±0.36 |
| **ResNet20** (Full Network) | 94.37 | - | - | 73.45 | - | - |
| LT [9] | 91.69±0.06 | **90.12±0.12** | **86.92±0.21** | 67.25±0.44 | 63.47±0.10 | 52.40±0.42 |
| SNIP [23] | 91.76±0.06 | 89.91±0.07 | 85.28±0.07 | 67.21±0.38 | 61.88±0.23 | 47.25±0.77 |
| GraSP [41] | 91.64±0.17 | **90.24±0.15** | 86.60±0.05 | 67.53±0.46 | **63.60±0.06** | 53.82±0.38 |
| Random Tickets (Ours) | **91.88±0.01** | 90.13±0.10 | 86.66±0.09 | 67.49±0.35 | 63.42±0.18 | **54.62±0.32** |
| **ResNet56** (Full Network) | 94.49 | - | - | 76.94 | - | - |
| LT [9] | 92.70±0.42 | **92.14±0.23** | **89.59±0.40** | 71.50±0.44 | 69.11±0.43 | 54.24±0.56 |
| SNIP [23] | 93.37±0.11 | 43.05±19.67 | *10.00±0.00* | 63.01±4.15 | 6.52±3.86 | *1.00±0.00* |
| GraSP [41] | 93.30±0.15 | **92.22±0.10** | 22.14±13.56 | **73.64±0.19** | 70.03±0.51 | 10.50±9.22 |
| Random Tickets (Ours) | **93.53±0.11** | 92.24±0.21 | 89.80±0.24 | 72.87±0.66 | **70.88±0.39** | **63.60±0.58** |

Table 7: Test accuracy of partially-trained tickets and our hybrid tickets of VGG11/16 and ResNet20/56 on CIFAR-10 and CIFAR-100 datasets.

| Dataset | CIFAR-10 | | | CIFAR-100 | | |
|---|---|---|---|---|---|---|
| Pruning ratio | 90% | 95% | 98% | 90% | 95% | 98% |
| **VGG11** (Full Network) | 92.05 | - | - | 69.09 | - | - |
| Learning Rate Rewinding [37] | **92.23±0.07** | **91.81± 0.22** | 88.71±0.36 | 69.41±0.29 | 68.70±0.25 | 66.64±0.36 |
| Hybrid Tickets (Ours) | 92.21±0.13 | 91.61± 0.12 | 91.04±0.21 | 69.60±0.34 | 69.08±0.41 | 67.45±0.02 |
| **VGG16** (Full Network) | 93.77 | - | - | 73.24 | - | - |
| Learning Rate Rewinding [37] | **94.01±0.12** | 93.70±0.22 | *10.00±0.00* | **74.14±0.17** | 72.82±0.35 | 65.51±1.41 |
| Hybrid Tickets (Ours) | **94.01±0.11** | 93.89±0.22 | 93.31±0.04 | 73.81±0.28 | 72.88±0.44 | 70.75±0.15 |
| **ResNet20** (Full Network) | 94.37 | - | - | 73.45 | - | - |
| Learning Rate Rewinding [37] | **93.46±0.15** | 92.12±0.14 | 88.78±0.23 | **70.45±0.10** | 65.27±1.26 | 54.77±0.55 |
| Hybrid Tickets (Ours) | 93.18±0.19 | 92.13±0.12 | 89.06±0.17 | 68.83±0.31 | 65.77±0.84 | 57.74±0.59 |
| **ResNet56** (Full Network) | 94.49 | - | - | 76.94 | - | - |
| Learning Rate Rewinding [37] | **94.57±0.34** | 94.13±0.26 | 92.35±0.14 | 75.69±0.71 | 68.22±0.46 | 57.82±0.48 |
| Hybrid Tickets (Ours) | 94.10±0.34 | 93.39±0.13 | 92.02±0.13 | 73.37±1.12 | 71.20±0.90 | 65.44±1.70 |

# D   Discussions of Iterative Magnitude Pruning (IMP)

In [9], the authors propose to find initial tickets with *extremely high sparsities* by iteratively throwing out weights with the smallest magnitude since one-shot LT method can find initial tickets with a certain layer being fully pruned. This method is named as iterative magnitude pruning (IMP) and is regarded as a standard pruning method such as in [37] and [10]. In this section, however, we show that the iterative procedure itself **cannot** help initial tickets to pass our sanity-checks although it can help these tickets to be trainable at a high level of sparsity.

In both [9] and [27], the authors admit that initial tickets found in VGG and ResNet architectures with CIFAR-10 and more complicated datasets are robust to re-initialization, and thus is not efficient. However, since this phenomenon is verticle to the property we discuss throughout our paper, i.e., whether the iterative procedure can help to find a good structure or to encode data information into

initial tickets, for the integrity of our work, we apply our sanity-checks on iteratively-found initial tickets and show the experiment results in Figure 5.

Figure 5: Sanity checks of iteratively-found initial tickets of ResNet32 and VGG19 on CIFAR10.

The results are similar to those of one-shot LT initial tickets, that is iteratively-found initial tickets cannot pass sanity-checks. One interesting thing we observe from the figure is that although on VGG the *corrupt data* check causes a performance gap, the tickets obtained by it is much more stable when the sparsity is high.[13] This suggests there is still room to explore how to use the data effectively to get trainable sparse initial tickets.

Moreover, we investigate the efficiency of iterative tickets including iterative LT and iterative LR rewinding. The results can be found in Table 8. From the table we can get the following several conclusions.

- **The iterative procedure even hardly helps when finding initial tickets of modern architectures as ResNet and VGG.** Compared to LT, Iterative LT only gains a little improvement or even performs much *worse* than LT. Furthermore, our Random Tickets can behave better than both Iterative LT and LT. This observation further strengthens our statement that initial tickets' performance to a large extent depend on the *layerwise keep-ratios* obtained. At the same time, this observation suggests that it may be difficult to answer the question of *whether we can find a kind of initial ticket that can pass sanity-checks.* The exploration will be left as future works.

- **The iterative procedure helps a LOT when finding partially-trained tickets.** Compared to Learning Rate Rewinding, Iterative LR rewinding can benefit from the iterative process and thus becomes a hard-to-beat sota up to now.

- **Iterative Hybrid Tickets can also benefit from the iterative procedure.** Due to the simple structure of our smart ratio, the Iterative Hybrid Tickets (IHT) cannot surpass the Iterative LR rewinding method. However, note that the performance gap is small, we assert that this phenomenon together with our sanity-checks on LR rewinding suggests there *must be some information encoded in both weights and partially-trained tickets' structure*, and that's the reason causes the different effectiveness of the iterative procedure on initial/partially-trained tickets. This sheds light that if we can find a better way to determine the layerwise keep-ratios explicitly, we can find a better pruning method instead of keeping weights of largest magnitude globally to implicitly find those keep-ratios.

# E    Sanity-Checks on More Complicated Dataset

Also for the integrity of our work, in this section we provide results of applying our sanity-checks using the CIFAR-100 dataset with ResNet32/VGG19 architectures on initial tickets. The results are presented in Figure 6.

Obviously, as we expect, we can see conclusions on initial tickets still hold on this more complicated dataset: initial tickets all failed to pass our sanity-checks, suggesting the *generalizability across datasets* of our checking methods.

Table 8: Test accuracy of pruned VGG19 and ResNet32 on CIFAR-10 and CIFAR-100 datasets, indicating the effectiveness of the iterative procedure on both LR Rewinding and Hybrid Tickets.

| Dataset | CIFAR-10 | | | CIFAR-100 | | |
|---|---|---|---|---|---|---|
| Pruning ratio | 90% | 95% | 98% | 90% | 95% | 98% |
| **VGG19** (Full Network) | 93.70 | - | - | 72.60 | - | - |
| LT [9] | 93.66±0.08 | 93.39±0.13 | *10.00±0.00* | 72.58±0.27 | 70.47±0.19 | *1.00±0.00* |
| Random Tickets (Ours) | 93.77±0.10 | 93.42±0.22 | 92.45±0.22 | 72.55±0.14 | 71.37±0.09 | 68.98±0.34 |
| Iterative LT [9] | 93.72±0.16 | 92.84±0.49 | 78.85±9.80 | 72.74±0.36 | 71.18±0.12 | 67.69±0.50 |
| Learning Rate Rewinding [37] | **94.14±0.17** | **93.99±0.15** | *10.00±0.00* | 73.73±0.18 | 72.39±0.40 | *1.00±0.00* |
| Iterative LR rewinding | **94.13±0.09** | **94.16±0.19** | **93.95±0.14** | **74.53±0.09** | **74.52±0.07** | **72.90±0.11** |
| Iterative Hybrid Tickets (Ours) | 93.95±0.12 | **94.15±0.20** | 93.87±0.24 | 74.24±0.04 | 74.19±0.06 | 72.63±0.28 |
| **ResNet32** (Full Network) | 94.62 | - | - | 74.57 | - | - |
| LT [9] | 92.61±0.19 | 91.37±0.28 | 88.92±0.49 | 69.63±0.26 | 66.48±0.13 | **60.22±0.60** |
| Random Tickets (Ours) | 92.97±0.05 | 91.60±0.26 | 89.10±0.33 | 69.70±0.48 | 66.82±0.12 | 60.11±0.16 |
| Iterative LT [9] | 92.46±0.15 | 91.22±0.36 | 88.44±0.19 | 69.13±0.21 | 66.69±0.45 | 59.53±0.57 |
| Learning Rate Rewinding [37] | 94.14±0.10 | 93.02±0.28 | 90.83±0.22 | 72.41±0.49 | 67.22±3.42 | 59.22±1.15 |
| Iterative LR rewinding | **94.86±0.07** | **94.71±0.08** | 93.43±0.15 | **74.57±0.08** | **72.59±0.12** | 67.17±0.72 |
| Iterative Hybrid Tickets (Ours) | 94.73±0.04 | 94.35±0.07 | **93.45±0.12** | 72.89±0.31 | 71.23±0.11 | **67.81±0.09** |

Figure 6: Sanity checks of the initial tickets of ResNet32 and VGG19 on CIFAR100.

# F  Other Related Works

We add some related works on network pruning. Most pruning methods are applied to pre-trained networks and thus *require training the full network* [33, 22, 15, 6]. Another line of pruning methods prune the models during training [28, 5, 46, 12]. However, these methods do not save many resources as they require the whole network during training time [17].

Towards a better understanding of network pruning, [27] evaluates several *structrued pruning methods* and concludes that training a large, over-parameterized model is often not necessary to obtain an efficient final model. [3] releases a library, ShrinkBench, for comprehensively evaluating pruning methods. [32] studies the transfer of pruned networks across datasets. [30] gives a theoretical understanding of the lottery tickets hypothesis. [45] shows the importance of setting weights to zero, the signs of weights and masking. [43] shows the tickets can be obtained in an earlier phase than that in [9].