[Reviews · NeurIPS 2020]

Review 1

Summary and Contributions: This paper proposes a new method of network pruning. The paper starts with two discoveries of network pruning 1. Previous methods only use limited information from training data 2. Different architectures of previous methods don't affect much of the result. This paper then proposes a zero-shot pruning method which has similar performance comparing to the previous methods, and by combining this pruning algorithm with previous method it could get more improvement.

Strengths: 1. The logic flow of this paper is clear. The sanity check is very convincing. 2. The finding that current pruning methods are not relying much on training data is surprising but reasonable. The proposed pruning method is much more efficient comparing to conventional pruning methods. 3. Emperical evaluation shows the proposed sanity check is crucial and necessary. 4. The relevance to the NeurIPS community is high.

Weaknesses: Hybrid tickets shows only marginally improvement in the result. The training process becomes more volatile when the pruning ratio is large, it's possible that results from a fixed learning rate can't reflect the whole picture. I believe the result from random tickets are impressive enough though. Minor stuffs: In 2.1 notation, "m_l is the number of weights in layer l" is inappropriate. Should be m_l is the size of the weight or number of neurons.

Correctness: Yes. The claims are back up with experimental evidence.

Clarity: Yes.

Relation to Prior Work: Yes, the proposed method oppose the assumption of most previous works.

Reproducibility: Yes

Additional Feedback: -----------------------------After rebuttal I don't think the rebuttal resolves my concern about hybrid ticket. But I will hold my score since I believe the result and analysis for random ticket is impressive enough.


Review 2

Summary and Contributions: This paper performs two sanity checks on unstructured pruning methods - does data or architecture matter for pruning - and expands on observations to create smart ratios and hybrid tickets. - The authors check the dependency on data by using random labels, random pixels, and less data. They find that good tickets can still be found on such corrupted data. - They also check the dependency on structure by rearranging masks or shuffling unmasked weights within each layer. They find that performance does not drop when the per-layer structure is disturbed. - Inspired by these observations, they propose smart-ratios, where random tickets with the given smart-ratios mostly match or outperform other existing pruning methods. - They combine smart-ratios with learning rate rewinding to form "hybrid tickets" to gain further performance improvements.

Strengths: Relevance: pruning methods, particularly lottery tickets, have had growing interest recently. While the findings in this paper are not groundbreaking, I think it can be a valuable contribution to the ongoing research on pruning. Novelty: this paper builds on existing pruning methods, but provides some new insight on them and uses those insights to create more efficient variations.

Weaknesses: Soundness/significance: My main concern is that this paper focuses on one shot pruning rather than iterative pruning. Iterative magnitude pruning (IMP) is the standard in lottery ticket training since it does better, especially at higher prune rates. Even "Learning both Weights and Connections for Efficient Neural Networks", referenced in this paper for one-shot pruning, said iterative pruning is better. Do the sanity checks hold up on IMP? Do smart-ratios and hybrid tickets still perform better than IMP baselines? If so, you could argue that your method saves even more compute; if not, then the scope of this paper is more limited. Other concerns about the soundness of claims: - Table 3: I don't see half dataset or shuffled weights doing worse than LT - Table 4: hybrid tickets don't seem much better besides at 98% sparsity when the baselines fail

Correctness: The empirical methods seem correct.

Clarity: The writing is clear, however, I don't agree that "data used in the pruning step and the structure of pruned network are essential" are good "sanity checks": it is not bad for pruning methods to not rely on data/architecture. In fact, the ideal pruning method would be efficient and not need any additional information. I would instead frame those "sanity checks" simply as characteristics of pruning methods, which are used as inspiration for more efficient methods.

Relation to Prior Work: Generally yes, but I would like to see discussion about "One ticket to win them all: generalizing lottery ticket initializations across datasets and optimizers" for its relevance on two important points: - Figure 1b shows the pruning ratios by layer, so the observation that "keep-ratios decrease with depth" is not novel. - Appendix A2 compares the winning ticket to "layerwise rearrange", which they call "locally permute mask", and find that lottery tickets actually do significantly better than layerwise rearrange at high pruning ratios. (Will figure 3 in this paper show the same results if it were extended further right and used log scale to show the high pruning ratios more clearly?)

Reproducibility: Yes

Additional Feedback: See comments under weaknesses, clarity, and prior work. I would like to see broader experiments that consider IMP, clarification of claims from seemingly insignificant results in the tables, and a reframing of the paper that does not make it sound like pruning methods need to "pass" the tests of depending on data/architecture. I would also like to see "sanity checks" performed on datasets besides cifar10. I am surprised that corrupted data (figure 3) does not hurt performance at all - does the model end up with the same keep-ratios with corrupted data? Does the model need to train a lot longer for the corrupted data to have any effect (as per https://arxiv.org/abs/1611.03530)? Might be interesting to see more here. Minor questions/suggestions: - How many runs are used for error bars in figure 3 and the tables? - How close are the smart-ratios compared to actual keep-ratios from existing methods? - Figure 2 is good, would be better if all arrows/transitions had labels - This paper https://arxiv.org/abs/2006.05467 came out after the submission deadline, but it appears to be relevant (pruning without data, behavior of pruned layers); you might want to cite it in an updated version. This paper is going in an interesting direction and I would be happy to increase my score if the major concerns are addressed. --------------------------- Updates after rebuttal: Since the authors addressed many of my concerns, I have increased my score from a 5 to a 6. One additional comment for my review + the rebuttal about rephrasing how the term "sanity check" is used: my concern does not just apply to one sentence, but for the framing throughout the whole paper. Since a sanity check used to evaluate whether something is valid and meets certain requirements, it is fine to say "sanity checks on the beliefs that pruning relies on data and architecture", but it is not okay to say "sanity checks that pruning methods need to rely on data/architecture". I think this is important so that the paper doesn't imply that it's bad for pruning methods to not rely on data/architecture and thus "fail" checks. Examples where this is phrased well: L5, L37-38, L118. Examples of misleading phrasing: L16, 51, 74, 168, 183, 240, 243, 266, 284, 291.


Review 3

Summary and Contributions: The paper discovers that the initial weights and the architectures of lottery tickets found by pruning methods are not important for training a sparse neural network from scratch. What's important is the layer-wise sparsity ratio only, which should increases as the layer goes deeper. The discovery is very interesting and almost enables training sparse neural networks from scratch without pruning cost, if the conclusion can generalize.

Strengths: 1. a very interesting, inspiring and useful discovery: very sparse neural networks can be trained from scratch if the sparsity is randomly initialized in a way such that the sparsity increases as the layer goes deeper. 2. a certain settings of experiments to evaluate the discovery.

Weaknesses: 1. It's surprising that no large-scale results (i.e. ImageNet) are shown in the paper. Some previous works were not able to show large-scale results with the excuse of inefficiency of their pruning methods (although there are lots of efficient pruning methods out there), however, I believe the method proposed in this paper is efficient and should be able to run in ImageNet easily. I suspect the conclusion doesn't hold for large-scale problems. 2. The smart-ratio random tickets are concluded in a limited number of neural architectures/datasets, and tested in the same settings. The conclusion may not generalize to other settings. Such as, the sparsity can decrease/remain as the layer goes deeper as shown in Figure 4 [A]. More architectures/datasets should be used as meta-validation setting to validate the generalizability of the method. 3. The hybrid tickets only improve for Cifar-100 not for Cifar-10 in Table 4. [A] Jongsoo, et al. "Faster cnns with direct sparse convolutions and guided pruning." arXiv preprint arXiv:1608.01409 (2016).

Correctness: The experiments in the paper are correct, while it's unclear if the conclusion can generalize.

Clarity: 0. Consider to highlight the smart-ratio random tickets at the beginning. 1. "Second, it is believed that the architecture of the pruned network is crucial for obtaining the final pruned network with good performance [12]." More related works or experiments should be included to support this claim. Point out to me where [12] supports this claim. 2. I find "Fail to Pass Sanity Checks" and "partially-trained tickets can pass our sanity checks" is twisting to understand. Consider to rephrase. 3. Explain how the validation accuracy is obtained under corrupted datasets ("Random labels") when evaluating the process of pruning. 4. In Figure 3, use thinner lines and zoom into regions of interest (higher sparsity region with a smaller range at y-axis). Otherwise, it will give readers an impression of hiding something. 5. report layer-wise sparsity to support the "Keep-ratios decrease with depth." 6. Clarify/double check the “unmasked”/"masked" of "the unmasked weights of the pruned subnetwork" and "the masked network" between Line 248-252.

Relation to Prior Work: 1. For "there are structured pruning methods [17, 25, 22, 18]", some earlier/seminal works on structured sparsity/pruning methods [A,B,C] should be included, besides later works. [A] Li, Hao, Asim Kadav, Igor Durdanovic, Hanan Samet, and Hans Peter Graf. "Pruning filters for efficient convnets." arXiv preprint arXiv:1608.08710 (2016). [B] Wen, Wei, Chunpeng Wu, Yandan Wang, Yiran Chen, and Hai Li. "Learning structured sparsity in deep neural networks." In Advances in neural information processing systems, pp. 2074-2082. 2016. [C] Alvarez, Jose M., and Mathieu Salzmann. "Learning the number of neurons in deep networks." In Advances in Neural Information Processing Systems, pp. 2270-2278. 2016.

Reproducibility: Yes

Additional Feedback: *** My final score remains as the rebuttal didn't address my concerns. One advice to the writing: the authors should focus more on the part of finding tickets without training. The sparsity improvement is just a by-product. ***

[Author Response · NeurIPS 2020]

We thank all reviewers for taking their time reading the paper and providing us with insightful comments and suggestions!

**To R1:** Thank you for appreciating our work! Here are our responses.

• Regarding "results from a fixed learning rate": There may be some misunderstandings. We do not use a fixed learning
rate for any of our experiments, and the schedule can be found in Appendix B.

**To R2:** Thanks for your very careful review. We will rephrase the sentence you mentioned and add citations and
discussions on the related works you provided. Here are our responses to your questions.

• Regarding the sanity check on Iterative Magnitude Pruning (IMP): That is a good question, we focus on one shot
pruning in this paper for the following reasons: (1) As suggested by the result of Figure 7 in [Rethinking the value
of network pruning, 23] , if choosing proper learning rate (0.1, also suggested by [Rewinding paper, 33] ), other than
0.01 in the LTH and [Learning both Weights and Connections for Efficient Neural Networks, 12] papers, iterative
pruning cannot consistently outperform one shot pruning, and even when iterative pruning is better, the gap is small; (2)
Compared to one shot pruning, iterative pruning is obviously very time-consuming, and since our goal of the paper is
propose sanity checks for pruning methods, we chose to test on the rather efficient one shot pruning method. However,
we followed the reviewer's suggestion and conduct experiments on IMP settings. The results show that LT with IMP
has similar performance as one-shot LT and cannot pass the sanity checks in most settings; while IMP can boost the
performance of learning rate rewinding, which can pass the sanity checks as expected. The smart-ratio random tickets
can still outperform LT with IMP in most settings; while the improvement of hybrid tickets with IMP upon learning rate
rewinding with IMP becomes slighter or vanish in some settings (please see "Common reply" in the bottom for more
discussion on hybrid tickets). We will add the discussions in the next version.

• Regarding "half dataset or shuffled weights doing no worse than LT": This is an understandable result that applying
sanity check can hurt the performance of LT with rewinding, but the performance will not drop to under naive LT since
even naive LT will keep similar performance after applying sanity checks.

• Regarding the sanity check besides CIFAR-10: Thanks for the suggestion. Based on the empirical observation on
CIFAR-10, we proposed "random tickets" method in the paper. And the success of random tickets, in turn, corroborates
that we can achieve the performance of initial tickets without data information and certain network connection pattern.
We conducted experiments of random tickets on several datasets and architectures, and we believe these results are
enough for illustrating our ideas. We conduct sanity checks on other datasets in the version if the reviewer still suggests.

• Regarding the related results in [One Ticket paper, 28] and the lack of higher sparsity results: Thanks for pointing out
the very related results, we will add discussion on the difference in the next version. For the high sparsity regime, we
find that initial tickets still cannot pass the sanity checks. The difference between our observation and the figure you
mentioned may come from the "late resetting" trick used in [One Ticket paper, 28] (See the description in Section 3.1 in
that paper), which is a weak version of rewinding. So it is not surprising to see that Figrue A1 in [One Ticket paper, 28]
show evidence that the method may pass the layerwise rearrange check.

• Regarding some minor questions: (1) Each data point shown in the table/figure is obtained by averaging of 3 runs
(we would like to refer you to our Appendix B to see the detailed settings). (2) The trend of smart-ratios is similar to
keep-ratios from existing methods, we would like to add a visualized comparison in the next version.

**To R3:** Thank you for the careful review and helpful suggestions on writing and related work! We will polish the paper
and add citations as you suggested. Here are our responses to your questions.

• Regarding the large-scale dataset: For sanity-checking existing pruning methods, we follow the setting of the original
papers [SNIP paper, 20; GraSP paper, 36] to conduct our experiments on CIFAR-10/100 and Tiny-ImageNet. But as the
reviewer suggests, compared to exsiting methods, it will be more efficient to apply our random ticket on datasets like
ImageNet, we will conduct experiments to test the performance of our method on ImageNet.

• Regarding the generalizability of smart-ratio random tickets: We did examine different experiments settings to test
the generalizability of smart-ratio random tickets. Please see Appendix C for more experiments on several different
ResNet/VGGNet architecture.

• Regarding the claim of "the architecture is crucial": In their original paper the pruning phase is described as "learning
which connections are important and removing the unimportant connections", which implies the architecture is crucial.

**Common reply:** We notice all the reviewers pointed out the improvement of our Hybrid Tickets is not always significant.
We would like to point out that in this setting, the baseline (learning rate rewinding) itself is very strong, the performance
of networks pruned by this method can be even better than the original network. So we can expect that the room for
improvement is limited. However, our hybrid tickets still show strengths especially in the "hard settings" with hard
dataset (CIFAR-100) and high sparsity.

[Meta-Review · NeurIPS 2020]

This paper proposes a series of simple sanity checks to evaluate whether winning lottery tickets depend on information from the training data and whether the layerwise pruning statistics account for improvements of winning tickets. Finding that many pruning methods don't pass these two "sanity checks", the authors propose a series of data-independent pruning ratios which work across a number of settings. Overall, reviewers found the paper to generally be well-executed and the results to be interesting, though there were some concerns about the lack of iterative pruning and a lack of large-scale results. In the rebuttal, the authors described the results of their experiments with iterative pruning and promised to perform a large-scale analysis of their data-independent ratios. Reviewers were generally satisfied with these results, but I would strongly encourage the authors to include these experiments (with plots, which were lacking from the rebuttal) in the final paper. I therefore recommend that this paper should be accepted as a poster.